

# On the mechanical behaviour of a low angle normal fault: the Altotiberina fault (Northern Apennines, Italy) system case study

Luigi Vadacca[1], Emanuele Casarotti[2], Lauro Chiaraluce[2], Massimo Cocco[2]

[1]MOX, Dipartimento di Matematica "F. Brioschi", Politecnico di Milano, Milan, Italy
[2]Istituto Nazionale di Geofisica e Vulcanologia, Rome, Italy

*Correspondence to*: Luigi Vadacca (luigi.vadacca@polimi.it)

**Abstract.** Geological and seismological observations have been used to parameterize 2D numerical models to simulate the interseismic deformation of a complex extensional fault system located in the Northern Apennines (Italy). The geological

system is dominated by the presence of the Altotiberina fault (ATF), a large (60 km along strike) low-angle normal fault 20° dipping in the brittle crust (0-15 km). The ATF is currently interested by a high and constant rate of microseismic activity and no moderate-to-large magnitude earthquakes have been associated to it for the past 1000 years. Modelling results have been compared with GPS data in order to understand the mechanical behaviour of this fault and a suite of minor syn- and antithetic normal fault segments located in the main fault hanging-wall.

The results of the simulations demonstrate the active role played by the Altotiberina fault in accommodating the on going tectonic extension in this sector of the chain. The GPS velocity profile constructed through the fault system cannot be explained without including the ATF's contribution to deformation, indicating that this fault although misoriented has to be considered tectonically active and with a creeping behaviour below 5 km of depth.

The low angle normal fault also shows a high degree of tectonic coupling with its main antithetic fault (the Gubbio fault)

suggesting that creeping along the ATF may control the observed strain localization and the pattern of microseismic activity.

**Keywords.** Low-angle normal fault, Altotiberina fault, Northern Apennines, 2D numerical models.

## 1 Introduction

The mechanical behaviour of low-angle normal faults (LANFs; dip angle <30°) is a critical issue in fault mechanics (Jackson and White 1989, Buck 1993, Westaway 1999, Collettini and Sibson 2001, Collettini 2011) and it represents a paradox if the

paradigm of faulting in a brittle, elastic, homogenous crust holds. Frictional fault reactivation theory (Anderson, 1951; Sibson 1985) predicts that in extensional settings the maximum principal stress is vertical and no motion is possible on faults dipping less than 30° and sliding with a friction coefficient ($\mu_s$) in the range of 0.6–0.85 (Byerlee, 1978). This theory is supported by the evidence that no moderate-to-large magnitude earthquakes have been documented worldwide nucleating on





LANFs using positively discriminated slip planes from the focal mechanisms (Jackson and White 1989, Collettini and Sibson 2001). On the contrary, observations of large displacements (Lister and Davis, 1989; John and Foster, 1993; Hayman et al., 2003; Collettini and Holdsworth, 2004; Jolivet et al., 2010; Mirabella et al., 2011), and the well-identified microseismic activity (Chiaraluce et al., 2007; Rietbrock et al., 1996) associated with these structures suggest that the

LANFs are tectonically active accommodating crustal extension and possibly formed at low angle. For these reasons, the LANFs cannot be excluded from the assessment of seismic hazard, although their inclusion still represents a debated issue.

The LANFs mechanical paradox could be solved if one of the two basic hypotheses of the Anderson's theory is not verified: that is, if the orientation of the maximum principal stress on the fault plane is not vertical and/or if $\mu_s$ is much smaller than 0.6. Laboratory experiments on fault rock samples indicate that friction drops to very small values ($\mu_s \approx 0.2$) at high sliding

velocity suggesting that large portions of crustal faults are significantly weak (e.g. San Andreas Fault; Zoback et al., 1987). Moreover, depending on the amount of clay minerals in the fault core, the coefficient of friction may be significantly lower than 0.6 (Saffer et al., 2001; Brown et al., 2003; Collettini et al., 2009a; Lockner et al., 2011). Laboratory experiments on slipping zones rocks sampled on exhumed low-angle normal fault zones (e.g. Zuccale fault, Italy), have shown very low values of sliding friction coefficients, thus a prevalent velocity strengthening behaviour of the fault (Smith and Faulkner,

2010; Collettini et al; 2009b). For these reasons, some authors proposed that a stable sliding regime might characterize the LANFs behaviour (Chiaraluce et al., 2007; Hreinsdottir and Bennett, 2009; Collettini, 2011). If this frictional behaviour would be a common characteristic for the majority of such misoriented structures, it would explain the lack of moderate-to-large magnitude earthquakes observed to nucleate on these tectonic structures.

The frictional behaviour of faults is usually modelled in terms of rate- and state-dependent constitutive laws (Dieterich,

1994) in which friction depends on slip rate, state variables and normal stress. Spatial variations of frictional properties can explain the heterogeneity of crustal faulting and the frictional response to tectonic loading (both permanent and transient changes in loading conditions, Boatwright and Cocco, 1996). Velocity weakening and velocity strengthening frictional regimes are commonly used to identify dynamic instabilities (i.e., earthquakes) and stable sliding (aseismic or creeping), respectively.

In this paper we integrate geological, seismological and geodetic data with 2D numerical simulations of faulting to model the state of stress and strain of an active normal fault system located in the Northern Apennines of Italy. The system is dominated at depth by the presence of a large low angle normal fault (dip angle 15°-20°), named the Altotiberina fault (ATF; Barchi et al. 1998; Boncio et al. 2000; Figure 1). The 2D model is constrained by the geological and structural cross section published by Mirabella et al. (2011) based on the interpretation of seismic profiles, deep boreholes data and geological

observations at the Earth surface. We will use and interpret the on-going deformation inferred from GPS to understand the tectonic role played by this LANF and to study the effects of the interseismic tectonic loading on the state of stress and deformation of the whole ATF system together with possible strain partitioning and coupling in between the distinct active faults segments.



## 2 The Altotiberina fault system

The investigated fault system is located at the Tuscany-Umbria-Marche regions boundary within the Northern Apennines (Figure 1a), a NE-verging thrust-fold belt under-going NE-trending extension at a rate of about 3 mm/yr (Serpelloni et al. 2005). The system is dominated by a low angle normal fault: a ≈ 60 km long and 40 km width NNW-trending fault plane

dipping at 15°-20° toward ENE named Altotiberina fault (ATF; see fault trace at surface in Figure 1a). The depth range of the whole fault system is 0-15 km (Figure 1b).

In the ATF hanging-wall block, synthetic and antithetic structures dipping at higher angles have generated moderate magnitude earthquakes; the largest one was a $M_W$ 5.1 occurred in 1984, currently named the Gubbio earthquake (Figure 1a; Westaway et al., 1989; Collettini e al., 2003). Several authors suggest that this seismic event, originally named as the Perugia

earthquake (Haessler et al., 1988), has nucleated on the Gubbio fault plane (GuF; Haessler et al., 1988; Pucci et al. 2003) while Collettini et al. (2003) by interpreting a set of multidisciplinary data, argued against this hypothesis.

Following Chiaraluce et al., 2007, only micro seismic events (< 2.3 $M_L$) have been located along a 500-1000 m thick fault zone cross cutting the upper crust from 4-5 km down to 14-16 km depth and coinciding with the geometry and location of the ATF (Figure 1b) as derived from geological observations and interpretation of depth-converted seismic reflection

profiles (Mirabella et al., 2011). While in the hanging wall of the ATF the distribution of earthquakes highlights the presence of higher angle (40°-60° of dip) synthetic and antithetic minor normal faults (4–5 km long) that sole into the detachment. The seismicity nucleating along the ATF is characterized by a nearly constant rate of earthquake production r = 7.30e−04 eqks/day*km$^2$ (Chiaraluce et al. 2009), corresponding at about 3 events per day with $M_L$< 2.3. This seismic activity is cinematically consistent with the local tectonic setting characterized by the ATF low-angle plane and shows a higher $b$-value

than the seismicity located in the hanging-wall block (antithetic structures) showing seismic sequences like behaviour. Chiaraluce et al., (2007) interpreted these features as the presence within the fault system of fault zones possessing different rheology and/or frictional properties. It is also worth noting that the microseismicity nucleating on the ATF is not able to explain the amount of deformation associated with the short and long-term slip rate inferred by geological (Collettini and Holdsworth 2004) and geodetic studies and data (D'Agostino et al. 2009). These observations together with the lack of large-

magnitude (M > 7) historical earthquakes that ruptured the whole ATF in the past 1000 years (Rovida et al. 2011, Chiaraluce et al., 2014) suggest the occurrence of aseismic deformation or creeping as proposed by Hreinsdóttir and Bennett (2009) by interpreting regional GPS data. This hypothesis is supported by laboratory experiments performed on fault rock samples of the Zuccale low-angle normal fault considered the (older) exhumed analogue of the ATF. Talc minerals, characterised by a very low friction coefficient over a wide range of environmental conditions (0.05< μs<0.23; Moore and Lockner, 2008), has

been in fact observed to form interconnected foliated networks within the Zuccale fault core resulting in a velocity strengthening behaviour (Smith and Faulkner, 2010; Collettini et al, 2009b).





## 3 Numerical simulations

We perform 2D finite element numerical simulations with plain-strain approximation by means of the commercial software COMSOL Multiphysics (http://www.comsol.com/). In order to parameterize the numerical model, we use a NE-SW geological cross-section cutting the central part of the ATF system (Figure 1a) thus considering the whole set of associated

faults segments defined by Mirabella et al. (2011; see Figure 4b).

The mesh consists of approximately 270.000 triangular elements with a finest resolution of 25 m near the fault zones and decreases in resolution down to 2000 m along the boundaries. The crust is characterized entirely by an elastic rheology and we include those layers representing the main lithological units characterized by homogeneous competence (Pauselli and Federico, 2003; Mirabella et al., 2011; Table 1). We prolong the layers in order to avoid boundary effects during the

simulation. Hence in proximity of the boundaries, the layers are maintained flat because no information is available about their realistic setting (Figure 1b).

In order to facilitate the convergence of the solution, the simulations were performed in two subsequent stages. In the initial stage, the model was subject only to the gravity load; this means that no velocity at the boundaries was imposed to simulate extension. In this way, the model compacts under the weight of the rocks and it is brought in a stable equilibrium with

gravity. In this first step, the boundary conditions, applied to all models, are the following: (a) the upper boundary of the mesh is free to move in all directions; (b) the lateral boundaries of the mesh and the bottom are kept fixed in the perpendicular direction. This means that slip parallel to these boundaries is allowed. In the second stage (interseismic phase) we start stretching the model (e.g. crustal extension) for 10 years, applying a constant horizontal velocity of 0.5 mm/yr and 3.5 mm/yr on the SW and NE lateral boundaries, respectively, in according to the present-day strain rate and kinematics of

the region (Serpelloni et al., 2005). The stress field resulting from the first stage is defined as uniaxial strain reference frame following Engelder (1993). This state of stress is characterized by vertical stress

$$S_v = \rho g z \tag{1}$$

where ρ is the density, $g$ is the gravity acceleration and $z$ is the depth, and horizontal stress

$$S_H = \left(\frac{v}{1-v}\right) S_v \tag{2}$$

where ν is the Poisson's ratio. In this way for ν = 0.25, the vertical stress is three times larger then the horizontal stress, mimicking the extensional tectonic regime of the study area.

In the simulations the faults (see cartoon in Figure 2a) are defined as 100 m thick shear zones (Figure 2b). Only the ATF zone is represented by a thickness of 500 m as proposed by Chiaraluce et al. (2007). The weaker rheology of the fault zones is modelled by considering a variation of the elastic properties respect to the hosting intact rocks (e.g. Gudmundsson, 2004;

Faulkner et al., 2006). In this way, when a fault is considered locked the Young modulus of the corresponding fault zone is set to 10 GPa (Figure 2b) meaning that the deformation associated with that structure will depend mainly on the intensively fractured rocks typically, for instance, of the damage zone. Otherwise, when creeping faults are considered, the Young modulus of the corresponding fault zone is reduced by three orders of magnitude lower then the intact rocks (0.01 GPa, see



Figure 3). In this case the deformation associated with the causative fault will depend mainly on the creeping layer and on minor contributes from the damage zone. This abrupt mechanical contrast between the fault zones and the surrounding intact rocks along the hanging wall and footwall blocks simulates a slip similar to those observed when creep motion is modelled along fault planes. However, the adoption of the creeping fault equivalent method results in reaching more easily a numerical

convergence in solving the problem and a reduction of the computation costs.

Through the numerical simulations we aim to model the present day deformation of the ATF system (Figure 1b). Model simulations are then validated through their comparison with a 2D GPS displacement profile derived by the GPS data (Figure 1a and Figure 4a) retrieved by the network installed in the study area. The extraordinary density of the GPS network allows the calculation of high resolution 2D displacement profile through the fault system thank to the recent implementation

of the National GPS Network (http://ring.gm.ingv.it/) managed by the Istituto Nazionale di Geofisica e Vulcanologia (INGV) that also benefit from the implementation of a multidisciplinary research infrastructure, The Altotiberina Near Fault Observatory, devoted to the understanding of the physics of earthquakes and faulting (Chiaraluce et al., 2014). To retrieve the 2D displacement profile we use 11 GPS stations whose distance from the cross-section is less then 10 km (from SW to Ne they are: SIO1, REPI, CSSB, VALC, UMBE, ATLO, MVAL, PIET, ATFO, ATBU, FOSS; Figure 1a).

Thus, in order to quantify the interseismic stress build-up in the last 10 years we compute the Von Mises stress, a parameter that expresses the difference between the principal components of stress giving indication of the amount of shearing (e.g. Pauselli and Federico, 2003) and four different types of simulations have been performed (see Figure 3). In the first model all faults are considered locked (Model 1). In the second model (Model 2), creeping is simulated only along the ATF considering a (5 km) locking depth we derived from our tests (see next paragraph). In the third model (Model 3), we

simulate creeping only along the GuF, whereas the ATF is considered locked. While in the fourth model (Model 4A) creeping behaviour is simulated both along the ATF and GuF. This last model has been further complicated by considering within the ATF fault zone a set of multi-creeping-layers, as proposed by Chiaraluce et al. (2014).

### 3.1 Fault segments locking depth

In this section we discuss the effects of assuming different locking depths identifying the depth at which creeping begins

along synthetic and antithetic faults of the ATF system. The tests have been performed relying on the results of Vadacca (2014) and the following settings were considered for the analysis. First, we investigated the ATF locking depth by considering four characteristic depths whence the ATF is in creeping: 2, 5, 8 and 11 km, respectively. In this first test case, we do not consider the contribution of the other fault minor segments. In order to identify the optimal model parameterization, the model results are compared with the available GPS data (Figure 4a). The best fitting was obtained by

assuming an ATF locking depth of 5 km (Model-a in Fig. 4). In a second set of models, we considered the influence of modelling creeping along both the synthetic and antithetic faults of the ATF system by assuming different configurations for each of the ATF locking depths previously measured. Initially we considered that all the synthetic and antithetic faults were in creeping. Successively we locked these faults one-by-one from west to east (Figure 4b). The best fit for every model




setting is shown in Figure 4a. Models ATFb and ATFc correspond to an ATF locking depth of 2 and 5 km, respectively and creeping only along the Gubbio fault. Whereas the ATFd and ATFe models correspond to an ATF locking depth of 8 and 11 km, respectively and creeping simulated along all the synthetic and antithetic faults. From the analysis of the out coming results an ATF locking depth of 5 km leads to a better fit of the GPS data. This hypothesis is in agreement with the evidence

of microseismic activity occurring along the ATF plane starting from the same 4-5 km of depth (Chiaraluce et al. 2007) and the model resulting from regional geodetic data by Hreinsdottir and Bennett, 2009. In this best model we cannot exclude the creeping deformation interesting also a portion of the Gubbio fault. For this reason this model configuration will be analysed more in detail in the next section.

### 3.2 Modelling results

Figure 5 show the interseismic stress build-up for all the tested models represented in terms of Von Mises stress patterns. We observe that, when no creeping is included in the simulations (like in Model 1 of Fig. 3), the response of the medium to the imposed interseismic tectonic extension is completely controlled by the adopted lithology. In other words, the stress build up is only controlled by the mechanical contrast of the different lithology and the geometry of the fault system (Figure 5a). When creeping is simulated along the ATF, starting from 5 km of depth like in Model 2, the pattern of the stress build up

changes completely. Stress is mainly localized around 5 km of depth and in the ATF hanging wall (Figure 5b). This model also predicts a large stress accumulation along the deepest (3-5 km) portion of the GuF and low values of stress for the ATF footwall block where currently we observe a lack of seismic activity (see Fig. 1b). This deeper sector of the GuF shows a quite flat geometry contrary to its shallower portion where it is at high angle. Our model resolution does not allow separating the contribution of these two diverse fault portions even if in reason of its flat attitude, the deepest segment is the one being

more efficient. Coherently with this we observe that when we simulate creeping only along the GuF (Model 3), we observe that the stress pattern (see Figure 5c) ,is of course very similar to the reference model when no creeping is assumed (Model 1) in reason of the fault size. At the same time we note that the main differences are observed around 5 km of depth where on the GuF hanging wall (Gubbio basin) the values of stress decrease due to the continuous sliding along the fault. Moreover, higher values of stress are concentrated where the GuF changes the dip from lowest (3-5 km of depth) to highest (0-3 km)

dip values in reason of the higher slip-rate of the shallow fault portion.

The combined effect of creeping along ATF and GuF is finally simulated through the Model 4A and is shown in Fig. 5d. Here we observe a wedge of higher values of stress located between the ATF hanging wall and the GuF footwall in proximity of the assumed locking depth (5 km). This model predicts stress accumulation in the footwall of the GuF where micro-seismicity is currently observed (Valoroso et al., 2014). The last model differs from the previous one for the presence

along the ATF creeping sector of a fault zone composed by a multi-layer including alternate creeping and stick-slip layers (Model 4B in Fig. 3). The interseismic stress build-up decreases with respect to previous models and stress accumulation is mainly located at the intersection between the ATF and the GuF (Figure 5e).



Figure 5 provides a clear picture of the role played by the ATF presumably creeping along its deepest portion. At the same time we observe signatures of tectonic coupling between the ATF and the GuF antithetic fault in a context where the ATF is clearly dominating the mechanical behaviour of the whole system.

### 3.2.1 Stress build-up vs. fault displacement

Figure 6 shows the trend of the Von Mises stress and cumulative displacement computed across the ATF and the GuF fault, zones after 10 years of tectonic extension respectively at 9 and 3 km of depth. When no creeping is simulated (Model 1; pink dashed line in Fig. 6), the same stress threshold is reached on the hanging wall and footwall rocks both across the ATF (Figure 6a) and GuF (Figure 6b) zones. The Von Mises stress decreases into the fault zones due to the effects of more compliant elastic properties than the surrounding hanging-wall (HW-IR in Fig. 6) and footwall (FW-IR in Fig. 6) blocks

composed by intact rocks. No slip is generated across the fault zones because the faults are considered locked (Model 1 in Figure 6c and 6d). Otherwise, when creeping is simulated only along the ATF (Model 2; yellow dashed line in Fig. 6), the interseismic stress build-up decreases on the footwall rocks and into the fault zone but increase on the hanging wall rocks (Figure 6a). The stable sliding along the ATF generates slip rates of 1.5 mm/yr, thus resulting in a cumulative displacement that goes from 10 mm to 25 mm (Figure 6c). Even if the GuF is maintained locked, the creeping simulated along the ATF

affects the interseismic stress build-up on this structure. The stress tendency across the GuF is the same observed in Model 1 but the threshold is shifted towards higher values (Figure 6b). Even in this case no slip is generated along the GuF considered locked (Figure 6d). When creeping is simulated only along GuF (Model 3; light blue in Fig. 6), the trend of stress and displacement across the ATF is similar to that obtained for Model 1 but the interseismic stress build-up drops below values of 0.1 bar with higher values observed on the footwall respect to the hanging wall block (Figure 6b). It is notable that

in this case higher values of stress are located on the GuF footwall block while in Model 2, where only ATF was considered creeping, higher values of stress were detected on the ATF hanging wall block. This underlines that creeping faults can transfer tectonic stress in the hanging wall or footwall blocks only depending by their orientation. Slip rates along the GuF reach 0.5 mm/yr (Figure 6d). When both the ATF and the GuF are both considered creeping (Model 4A; red dashed line in Fig. 6) the stress threshold decreases across both the fault zones (Figure 6a and 6b). In this case higher values of stress are

detected in the GuF hanging wall block respect to the footwall (Figure 6b) due to the stable sliding motion along ATF loading the GuF hanging wall. Slip rates along the ATF and the GuF reach values of 2.2 and 1.5 mm/yr respectively (Figure 6c and 6d). Finally, when multi-creeping layers are simulated into the ATF fault zone (Model 4B; dark blue in Fig. 6) the interseismic stress build-up across the fault zone sharply decreases in both the fault zones (Figure 6a and 6b) with higher values on the hanging wall then the footwall block. Into the ATF fault zone are detected different layers with high and low

values of stress as consequence of the alternation of creeping and stick-slip layers (Figure 6a). This effect is evident also by the displacement behaviour across the ATF fault zone (Figure 6c). The slip-rates calculated increases to 2.8 and 2 mm/yr for ATF and GuF respectively.





Finally in Fig. 7 we show the comparison between the horizontal velocities computed for all the models described above and the observed GSP velocities. In order to compare the numerical results obtained in this study with the observations, the misfit is calculated through a weighted root mean squares (WRMS) approach. Model 1 and Model 3 do not show any relevant strain localization, contrarily to what the data suggest. Worthy of note, all the other three models (Model 2, 4A and

4B) predict strain localization coherently with the observations. The best fitting model is formally Model 4A (WRMS=0.19) displaying a well-defined velocity jump in proximity of 70 km on the cross section coherently with the observed GPS velocity. The effect of multi-creeping layers into the ATF zone (Model 4B) is increasing the velocity jump in the central part of the cross-section, although the calculated misfit (0.20) is very similar to that of the Model 4A.

## 4 Discussion

The results of the 2D simulations performed in this study have important implications on the role played by the ATF in accommodating the tectonic extension in this sector of the Northern Apennines. The first main outcome is that the GPS observations cannot be explained without considering the ATF contribution to deformation, indicating that this fault although misoriented has to be considered tectonically active. Our findings also indicate the need for creeping behaviour along the ATF below 5 km of depth At the same time our simulations suggest that using only the ATF does not exhaustively

explain the GPS data deserving the contribution of at least another active segment located in the ATF hanging-wall volume such as the Gubbio fault (Figure 7). In addition, the best fit of the GPS velocity data is obtained considering some degree of creeping also along a portion of the GuF. The mechanical behaviour of the west-dipping GuF is still debated in the literature (Boncio et al., 2000; Barchi et al., 1999; Mirabella et al., 2004, 2008; Barchi and Ciaccio, 2009). It is considered a seismogenic fault in active fault databases (*DISS Working Group Database of Individual Seismogenic Sources, version 3.1.0,*

*A compilation of potential sources for earthquakes larger than M 5.5 in Italy and surrounding areas, available at http://diss. rm.ingv.it/diss*) and the last moderate-magnitude event ($M_w$ 5.1; Westaway et al., 1989) occurred in 1984 in the area (namely, the Gubbio earthquake) occurred few kilometres apart (see location in Figure 1a). According to Collettini et al. (2003), the main shock has nucleated on a different fault (see location in the map of Fig. 1a). However, the spatial pattern of the micro-seismic activity of the Gubbio area is quite complex and it does not allow the identification of the GuF as a locked

active structure (Chiaraluce et al., 2007; Valoroso et al., 2014). We rely on the interpretation of the mechanical behaviour of the Gubbio fault as a mixed mode seismic/aseismic characterized by a relevant creeping component during the interseismic phase of the seismic cycle for both geometrical (flat attitude in the deeper portion and high-angle to ward the surface) and mechanical reasons. Our results are in fact in in agreement with the results of recent friction laboratory experiments performed on outcrop rocks from the Gubbio fault core (Finocchio et al., 2013). The experiments have shown that the

presence of layers with coupled velocity strengthening-velocity weakening behaviour (Bullock et al., 2014).

A dominating creeping behaviour of the ATF is in agreement with the lack of a large-magnitude historical earthquakes (M > 7) in the past 1000 years as well as with the results of laboratory experiments on rock samples of the Zuccale low-angle




normal fault (the western exhumed analogue of the ATF) suggesting low friction and a velocity strengthening behaviour (Smith and Faulkner, 2010; Collettini et al, 2009b).

Model-4A (fault zone with one creeping layer) and Model-4B (fault zone with multi creeping layers) present very similar values of weighted root mean squares (Figure 7). This complicates the interpretation of the structural setting and the mechanical behaviour of the ATF system. Complex fault zones with multiple strands are widely documented in strike slip regimes (e.g. Carboneras fault in southeastern Spain). These large fault zones present a distributed deformation over several metres of multiple active phyllosilicate-rich fault gouges. The derived mechanical behaviour is a prevalent creeping with small repeating earthquakes (Faulkner et al., 2003; Faulkner et al., 2008). However no evidence of multicores in exhumed low-angle normal fault zones are reported in literature (e.g. Collettini et al., 2011); for instance, a single-fault core zone, a few meters thick, characterizes the Zuccale fault. For this reason, a fault zone with a single core can be considered a plausible model for the ATF, where the stable sliding along this layer loads adjacent stick patches that can fail generating earthquakes.

The results of this study are preparatory for a 3D modelling of the Altotiberina fault system in which the lateral variations of frictional properties (Boatwright and Cocco, 1996) can be modelled to explain the observed seismicity pattern. Notwithstanding, the outcomes of these 2D simulations contain original indications and allow the definitive indication for an active LANF as well as for a tectonic coupling between the two major structures of this fault system generating a negligible stress accumulation in the ATF footwall, coherently with the lack of microseismicity, and a redistribution of the stress build up consistent with the high seismicity rate in hanging wall of the ATF, including the footwall of the GuF (Chiaraluce et al., 2007; Figure 1b).

## 5 Conclusions

In this work we integrate the results of 2D numerical models with geological, seismological and geodetic observations in order to understand some of the key features on the mechanical behaviour of the Altotiberina fault and its associated fault system. Our results show that:

- the tectonic loading of this sector of the Apenninic chain is mainly accommodated by and active LANF (the Altotiberina fault);

- such a LANF is creeping from about 5 km of depth;

- although the deformation caused by creeping along the Altotiberina fault is a first order condition to explain the strain concentration revealed by GPS data, a tectonically coupled model is needed to explain observations thus including a contribution of a partially creeping antithetic fault (the Gubbio fault);

- the stress redistribution within the fault system caused by creeping on both the ATF and GuF together with the presence of heterogeneous frictional properties on the ATF fault zone volume can explain the observed micro-seismicity pattern and the lack of moderate earthquake in the last 1000 years nucleating on this fault.



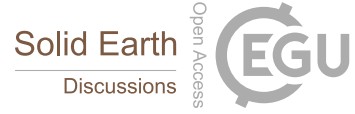

We speculate that the stable sliding along the creeping sections of the ATF may load adjacent stick patches within and around the fault zone that can fails in small seismic events. This mechanism, could explain the microseismic activity detected all along the ATF fault plane and within the ATF hanging wall.

### Acknowledgements

5 Spina Cianetti and Francesco Mirabella are thanked for scientific discussions on the numerical models and Umbria-Marche structural geology respectively. Enrico Serpelloni is thanked for GPS velocity data.

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

**Tables**

| | Young modulus (Pa) | Poissons ratio | Density (Kg/m$^3$) |
|---|---|---|---|
| Turbidites | 3.17e10 | 0.25 | 2390 |
| Carbonates | 6.68e10 | 0.25 | 2660 |
| Evaporites | 8.65e10 | 0.25 | 2800 |
| Pyllites | 5.33e10 | 0.25 | 2570 |
| Basament | 9.21e10 | 0.25 | 2840 |

**Table I**. Adopted parameters for the various rock formations (Pauselli and Federico, 2003; Mirabella et al., 2011).





**Figure Captions**

**Figure 1:** (a) Map of the study area located in the inner sector of the northern Apennines (see inset at the bottom). Grey points represent the epicentral location (from the catalogue of the INGV national network available at: http://iside.rm.ingv.it/iside/standard/index.jsp) of the earthquakes (M<3.5) occurred in between 1995-2010. The beach ball is the mechanism solution of the $M_w$ 5.1 1984 Gubbio earthquake

(Westaway et al., 1989; Collettini et al., 2003). The red lines represent the projection at surface of the Altotiberina faults system (after Mirabella et al., 2011). The yellow arrows represent GPS velocities vectors in the last 10 years (Vadacca et al., 2014). Blue line represents the cross-section used for the 2D models. (b) Cross section used for the numerical simulations on the basis of seismic profiles (after Mirabella et al., 2011). The grey points represent the micro-seismicity in a range of 10 km from the cross section (dashed blue line in Fig. 1a).

**Figure 2:** (a) Cartoon of the ATF faults system highlighting the ATF and Guf fault segments. Rectangle showing the zoomed area of Fig. 2b. (b) Sketch of the ATF and GuF faults zones where different fault mechanical behaviours are modelled using different Young modulus parameters and fault zone thickness (see text for details).

**Figure 3:** The different models configuration used to compute the interseismic stress build-up. First line: schematic sketch of the ATF (left column) and Guf fault zone (right column). The dashed areas represent the ATF and GF zones. The creeping deformation is modelled in the grey layers.

**Figure 4:** Horizontal velocity trend for different models obtained after the scouting of the ATF locking depth effects and creeping 20 simulated on different synthetic and antithetic faults (see Section 4 for details.). In the ATFa model a locking depth of 5 km is considered for the ATF fault and no creeping is simulated on the other faults; in the ATFb, ATFc models an ATF locking depth of 2 and 5 km is considered respectively and the creeping on the other minor faults is simulated only along the Gubbio fault, whereas in the ATFd and ATFe models an ATF locking depth of 8 and 11 km is considered and the creeping is simulated along all the synthetic and antithetic faults. We calculate the weighted root mean squares (WRMS) in order to compare the results obtained from the numerical simulations with the 25 GPS velocity observed: $WRMS_{Model-a}$= 0.30; $WRMS_{Model-b}$= 0.37; $WRMS_{Model-c}$= 0.19; $WRMS_{Model-d}$= 0.21; $WRMS_{Model-e}$= 0.23. The best fit is obtained for the Model-c.

**Figure 5:** 2D cross sections of the Von Mises stress distribution derived from the different models.

**Figure 6:** Trend of the Von Mises stress and cumulative displacement across the ATF (a-c) and GuF (b-d) fault zones after 10 year of tectonic extension and for the different adopted Models setting. The cross sections cut the fault zones at 9 km of depth for ATF and 3 km of depth for GuF. FW-IR: footwall intact rocks; HW-IR: hanging wall intact rocks. The thick grey lines define the ATF and Guf fault zones.

**Figure 7:** Horizontal velocity trend obtained for the different models setting. We calculate the weighted root mean squares (WRMS) in order to compare the results obtained from the numerical simulations with the GPS velocity observed. $WRMS_{Model1}$= 0.85; $WRMS_{Model2}$= 0.30; $WRMS_{Model3}$= 0.53; $WRMS_{Model4A}$= 0.19; $WRMS_{Model4B}$= 0.20. The best fit is obtained for the Model 4A.



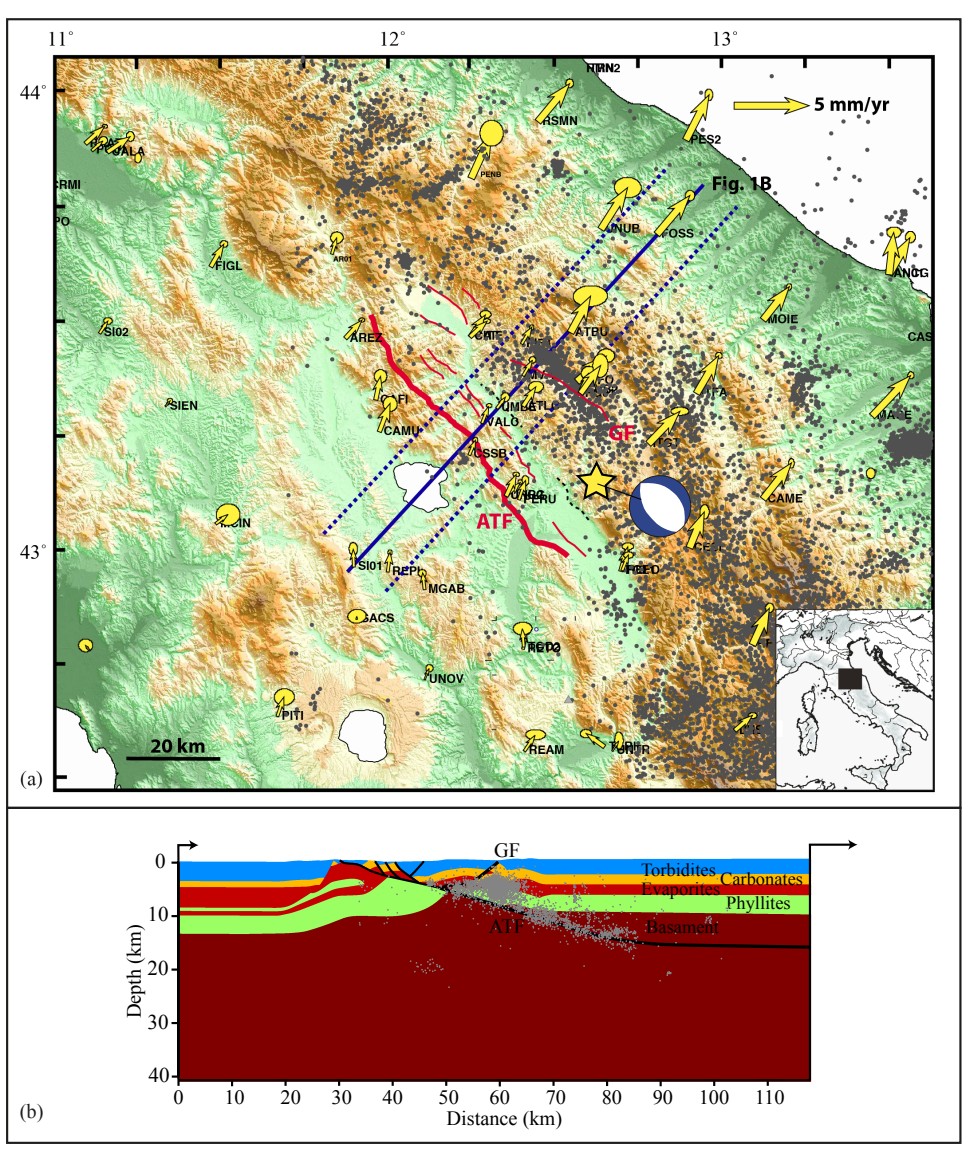

**Figure 1**



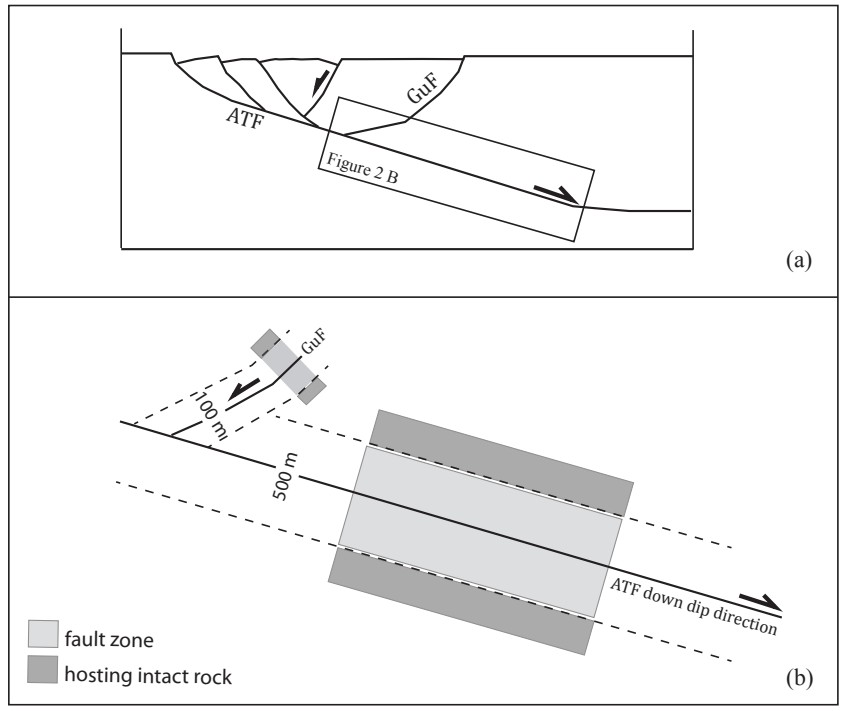

Figure 2



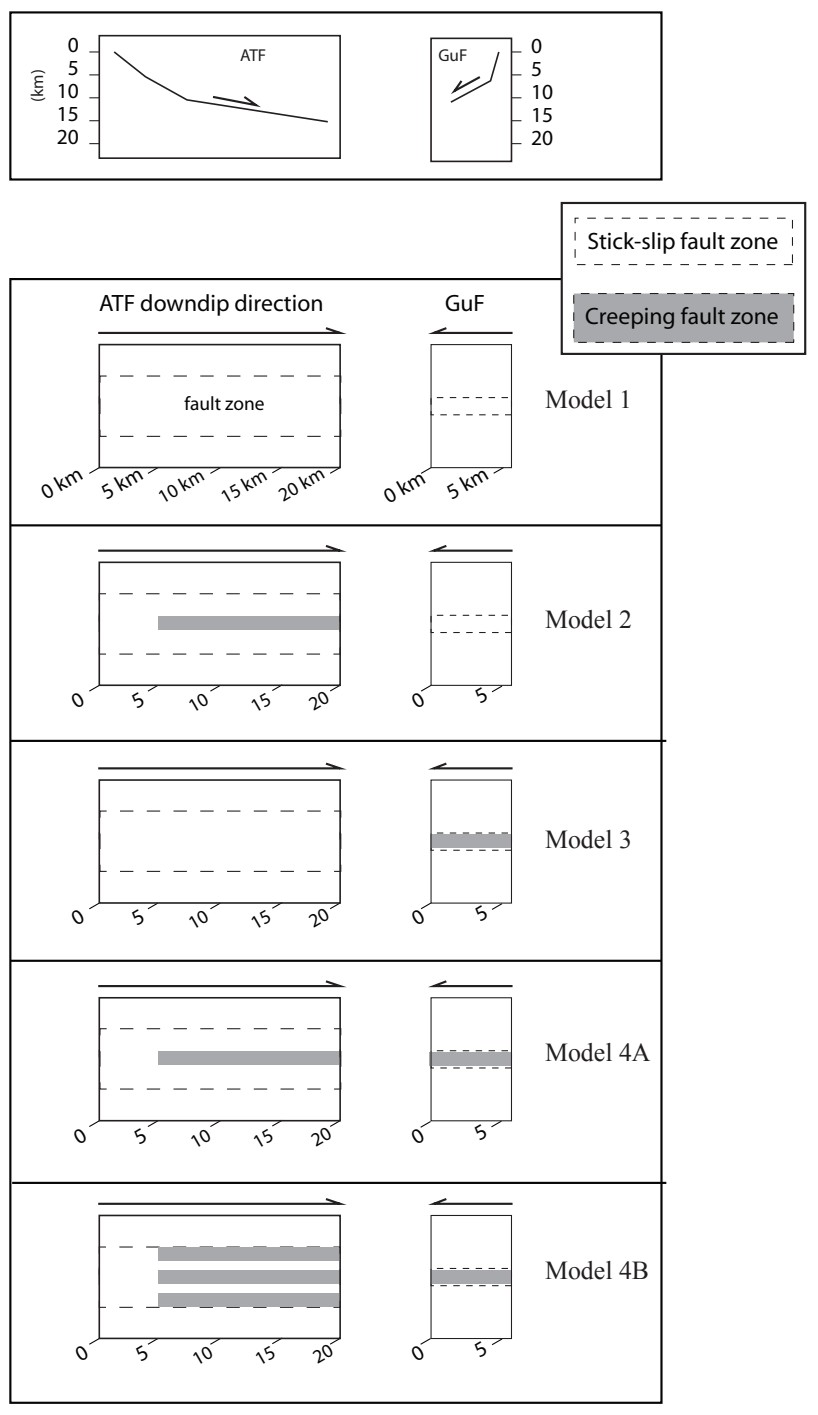

**Figure 3**





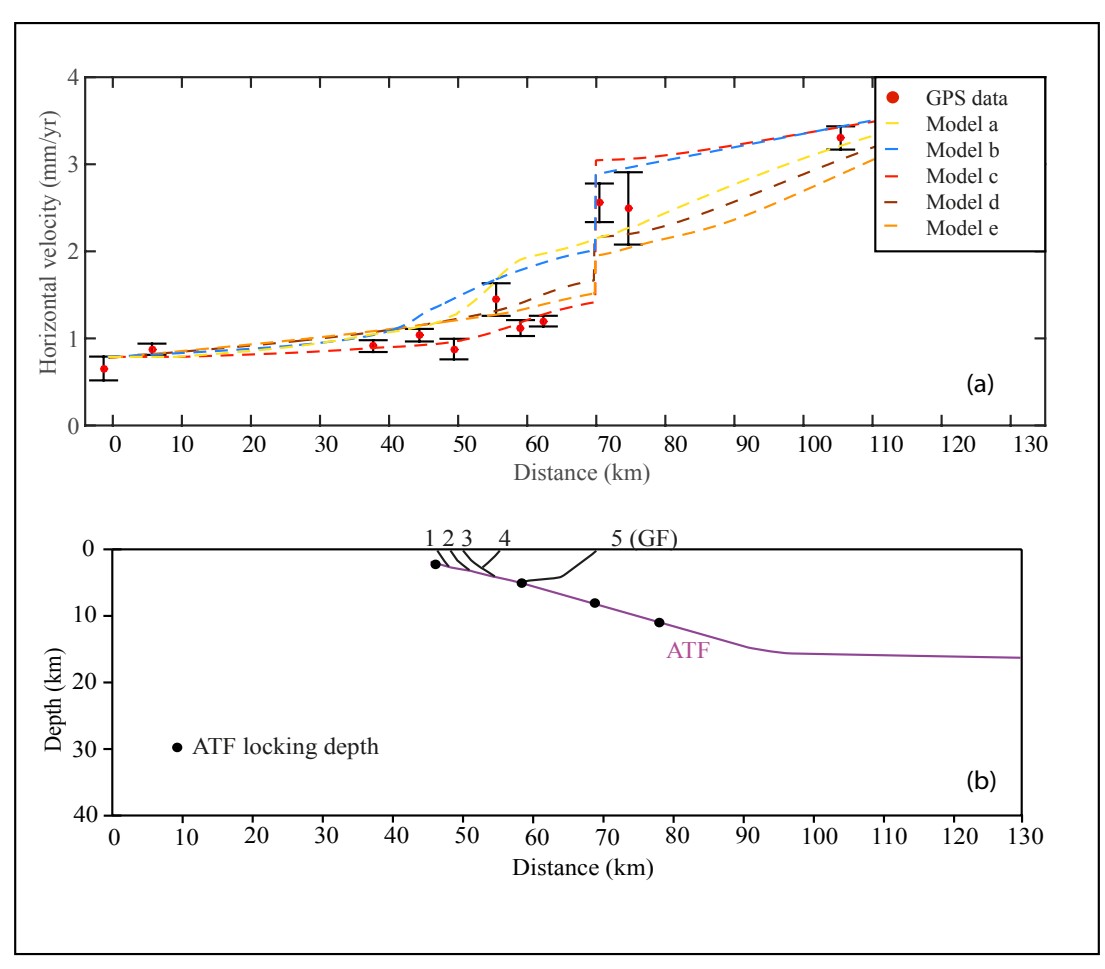

Figure 4





**Figure 5**




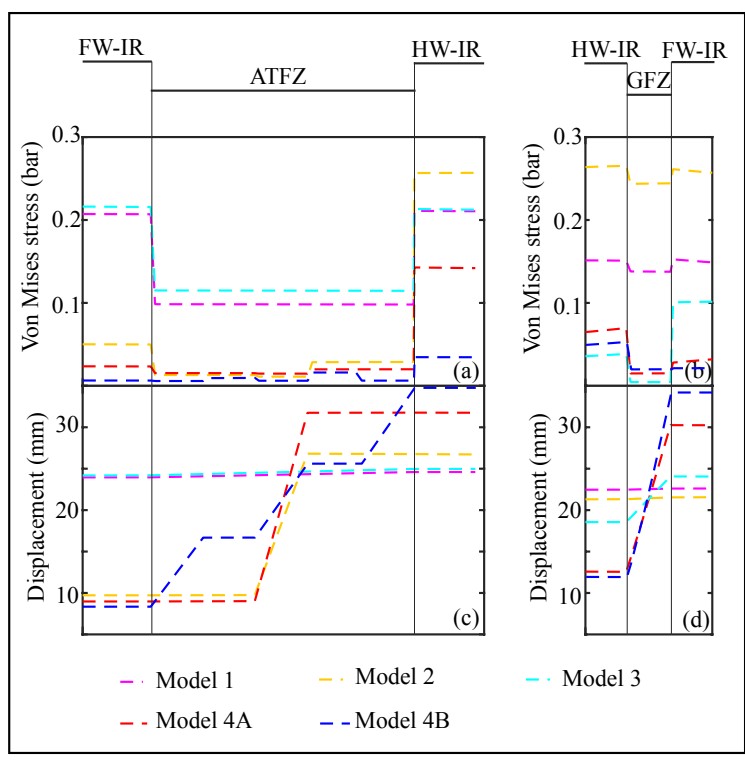

**Figure 6**





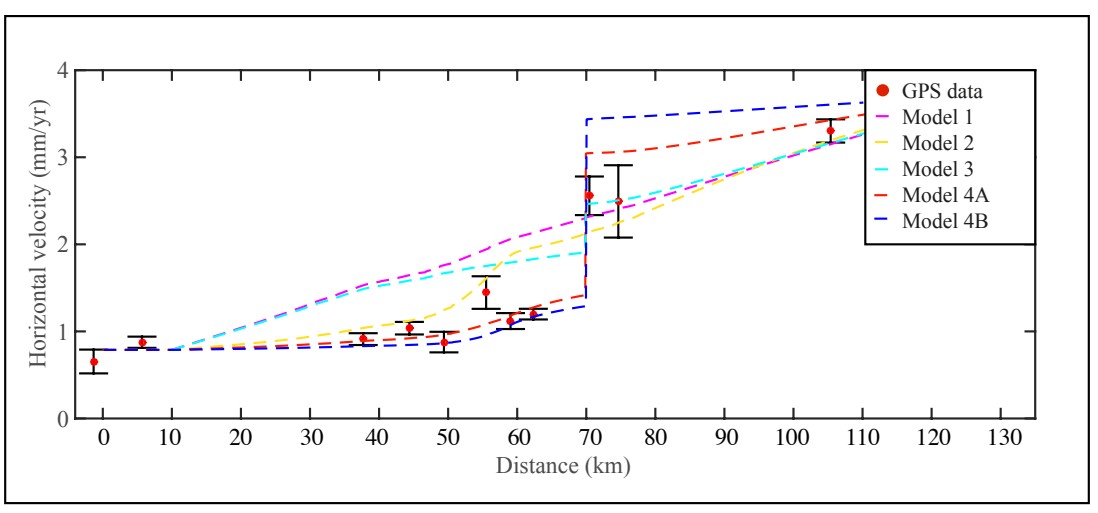

**Figure 7**