# Peer review of "On the mechanical behaviour of a low angle normal fault: the Altotiberina fault (Northern Apennines, Italy) system case study"

_Solid Earth, 2016_

## Short Comment (SC1) · 22 Mar 2016

I find noteworthy and promising the application of this kind of numerical modelling to the Alto Tiberina Fault system that in the past years has been massively studied by means of more geological and phenomenological, approaches. The joint exploitation of these two "investigative philosophy" should allow for deeper and more robust insights on complex seismotectonic systems like this.

As the authors likely know, their work cannot be considered conclusive until a more detailed fully 3D analysis has been completed. Nevertheless, I agree that a 2D approach is a suitable intermediate step the results of which could constitute a robust working foundation for a more complex 3D analysis. In this respect, I think that, taking advantage of the relatively minor computational workload represented by a 2D approach, the authors could extend the present analysis trying and implementing a sort of sensitivity analysis on their 2D results. Actually, the main result of the work, namely that the GPS velocities in the studied area cannot be explained without including the ATF contribution to deformation, is pretty robust and somewhat granted just looking at the peculiar GPS velocity field. Nevertheless, under what circumstances this conclusion could be weakened? Just given the robustness of their conclusions, it will be quite useful if the authors try and answer (from a numerical modelling point of view) to the previous question.

As a specific point, it is not clear to me the rationale of model 4B: what should be its geological/seismotectonical interpretation?

---

## Referee Comment (RC1) · B. Bennett (Referee) · 11 May 2016

General comments:

The paper by Vadacca et al "On the mechanical behaviour of a low angle normal fault: the Altotiberina fault (Northern Apennines, Italy) system case study" presents a 2D numerical model of crustal mechanics constrained by geological, seismological, and geodetic data. The objective of this modeling study is to characterize the mechanical behavior of the greater Alto Tiberina fault (ATF) system. The fault system is complex, involving interactions between the low-angle ATF and high angle faults that cut the ATF hanging wall and sole into the low angle normal fault. Understanding the mechanical behavior of this system is important because it bears on the general, controversial

question of if and how low-angle faults slip despite theoretical arguments to the contrary. The study also has important implications for local seismic hazards.

The numerical modeling suggests that the ATF plays an important role in controlling the ongoing extension across the fault system, and that the ATF appears to be creeping at depths below about 5 km. The results also suggest that creep on the ATF may have an important influence on some of the high angle normal faults, the Gubbio fault in particular, as well as micro seismicity.

Specific comments: The numerical model is developed using a finite element representation informed by seismic reflection lines. Creeping versus locked segments of the faults are simulated by changing the elastic properties of the fault zones relative to the surrounding material. The ATF fault is parameterized as a 500 m thick zone, whereas the other faults are parameterized as 100 m thick zones. Using the finite element model, a series of numerical experiments constrained by GPS observations of crustal motion were used to test the sensitivity of the observations to features of the model, such as the distribution of creep on both the ATF and high angle normal faults.

The results show that the accumulation of crustal stress depends critically on the prescribed fault creep. An analysis of predicted stress accumulation with the pattern of micro seismicity helps to inform which among the models tested best characterize the greater ATF system. An important outcome of the study is that models for which the ATF fault is creeping below about 5 km depth seem to fit the observations better that models that do not prescribe creep to the ATF. Further questions regarding creep on the Gubbio fault remain because the fit to the observations is similar for models in which both ATF and Gubbio faults creep; it is difficult to assess whether the small reduction in WRMS (change = 0.11) is achieved simply due to the increased complexity of the model. The same argument could be made to some extent for comparison among all of the models tested. However, it is notable that Model 4A is supported by the pattern of microseismicity, in that microseismicity correlates with high stress accumulation rates in the Gubbio fault footwall. The authors may be able to devise some statistical tests for

assessing the significance of differences in WRMS arising from models with different degrees of complexity.

Technical comments: The authors use the word "interested" in a way that I am unfamiliar with. Perhaps they meant "intersected"?

A table describing the models and reporting the WRMS values for each might help the reader to evaluate the differences among the models and their fit to the data.

A statistical assessment of the significance of differences in WRMS values among the various models would greatly strengthen the conclusions.

---

## Author Comment (AC1) · 25 May 2016

Dear Antonio,

thanks for your comments. I agree that 2D simulations can be considered an intermediate step versus the better understanding of the mechanical behavior of the Altotiberina fault. Nevertheless, as showed by Vadacca et al. (2014), current GPS data show little sensitivity to 3D features like fault topography. That is why we are working to create the opportunity to install additional GPS stations; probably the only way to provide fundamental progresses based on novel simulations considering 3D geological proprieties. However, for the aim of this work, that is to understand the "main" mechanical behavior of the ATF, 2D simulations can be considered exhaustive. This is also because there

are not large along-strike geometrical variations of the ATF plane (at least considering large distances), that could affect the orientation of the intermediate stress tensor.

Concerning your second question, the 4b model is devoted to verify the presence of layers of the fault zone with a different mechanical behavior as hypothesized by Chiaraluce et al (2014).

---

## Author Comment (AC2) · 1 Jun 2016

We appreciate the comments of the referee Rick Bennett on our study. They will help to improve our manuscript. We have written each reply just below to each comment.

1 - The authors use the word "interested" in a way that I am unfamiliar with. Perhaps they meant "intersected"?

P1, L11: We will change "interested" by "characterized"

P6, L7: We will also change "interesting" with "characterizing"

2 - A table describing the models and reporting the WRMS values for each might help

the reader to evaluate the differences among the models and their fit to the data.

We attach Table II (as supplement to this comment) reporting the WRMS calculations as suggested.

3 - A statistical assessment of the significance of differences in WRMS values among the various models would greatly strengthen the conclusions.

The relative uncertainty of root mean square error could be estimated as $(1/2A)^{1/2}$ with $A=sum(1/gpserror^2)$ under the assumption that the mean square error is (approximately) distributed as a $\chi2$ random variable. (Faber, 1999). In our case, the relative uncertainty is 5% that, for example, leads to a conservative absolute estimation for our minimum wrms of 0.01. In the paper, we report the uncertainty estimation to the wrms where needed. We highlight that we could consider wrms values of 0.19 and 0.24 as statistically different (95% confidence).

Nicolaas (Klaas) M. Faber, Estimating the uncertainty in estimates of root mean square error of prediction: application to determining the size of an adequate test set in multivariate calibration, Chemometrics and Intelligent Laboratory Systems, Volume 49, Issue 1, 6 September 1999, Pages 79-89, ISSN 0169-7439, http://dx.doi.org/10.1016/S0169-7439(99)00027-1.

Please also note the supplement to this comment:
http://www.solid-earth-discuss.net/se-2016-48/se-2016-48-AC2-supplement.pdf

**Supplement:**

| Model | Model_1 | Model_2 | Model_3 | Model_4A | Model_4B |
|-------|---------|---------|---------|----------|----------|
| WRMS | 0.85 | 0.30 | 0.53 | 0.19 | 0.20 |

**Table II.** Weighted root mean squares (WRMS) calculations for different models. In Model_1 all faults are locked; in Model_2 creeping is simulated only along the ATF considering a (5 km) locking depth; in Model_3 creeping is simulated only along the GuF, whereas the ATF is considered locked; in Model_4A creeping behaviour is simulated both along the ATF and GuF and finally in Model_4B multi-creeping-layers are simulated in the ATF zone.

---

## Referee Comment (RC2) · P. bernard (Referee) · 14 Jul 2016

This paper proposes to model the rather fast interseismic deformation in the region of the Alto Tiberina (ATF) fault using a 2D mechanical modl . The deformation rate around this major low angle normal fault, is provided by a dense cGPS array, and the detailed structure and fault geometries are provided by geological studies, deep boreholes, and avtive seismics. The authors use an elastic model for the whole crust, imposing horizontal extension at constant velocity. The model considers not only ATF but also smaller faults rooting in it, in particular the antithetic Gubbio fault . To numerically simulate the creep with the COMSOL software, they reduce the shear modulus to 0.01 GPa, in order to concentrate the elastic deformation close to the fault. They tried a

number of model, with varying locking depths of the various faults. Their conclusion is that for explaining the GPS records, the ATF must be creeping at depths larger than 5 km , and that the secondary faults play a significant role in the deformation by creeping. A second important result in the conclusion is that their model is consistent with the absence of reported moderate earthquakes in the last 1000 years.

General comment:

The topic is very interesting, and important, both for the mechanical problem of low angle normal faulting, and for the question of hazard assement related to the seismic potential of this major fault. The area is also the target of a Near-fault observatory, a densely monitored area, with multiparametric recordingÂǎ: this mechanical model is expected to provide a new frame for interpreting some of these data. However, the conclusions on the above questions are questionable: the purely elastic modeling may be leading to biased and possibly unrealistic results, and the inferred relatively moderate seismic potential of the studied faults may not be safely justified.

My main detailed comments are listed below.

(1) the introduction should quote published mechanical models of the crust with active faults, involving elasto-visco-plastic rheologies (e.g., Cianetti et al, GJI, 2008), and discuss the expected differences with their simplified, elastic model. Did the authors make some numerical comparison with simple fault/crustal modelsÂǎ to validate their simplified approach?

(2) in order to simulate creep on the major faults, the reduction by a factor of 1000 of the Young modulus results in a reduction by a factor of 1000 of the shear modulus (hence the creep), keeping the Poisson ratio constant (0.25). But this also reduces by a factor 1000 the incompressibility K: which means that not only the fault zone easily shears (mode II), but also easily compacts or expands (mode I). This may strongly alter the strain pattern outside the fault zones. This local mode I elastic strain is not a desired process (real creep should produce purely mode II slip), and may significantly
change the stress-strain transfers between neibouring faults, and with the more rigid blocks around. The authors do not mention nor quantify this expected side-effect of the model. I believe that a reasonable elastic simulation of creep should keep a large value of K in the fault zone, while decreasing significantly the shear modulus (by a factor of 10 or more). This can be done by adjusting the Poisson coefficient, taking it larger than 0.25. The relevance and optimization of such parametrization should then be tested in simple geometric, with the criteria of a neglectable mode I strain component on the fault zone - and numerical stability.

(3) A few locking depths have been tried, as shown in Figure 4 and 6. The main, sharp step between 65 km and 70 km varies strongly, depending on the model: Some models produce a larger step than the reported one from GPS, and the other produce a smaller step. Surprisingly, no model is presented with a proper adjustment of this main feature of the GPS record. I would imagine that some models with intermediate parameters would do the job (intermediate locking depth, and some active secondary faults) would provide a better adjustment to the reported step; but the authors do not mention this possibility: did they try?

(4) The Figure 1 shows that the deep microseismicity coinciding with ATF is not shallower than about 9 km in depth. The shallower seismicity appears off-fault, and close the Gubbio fault. The authors should comment on this apparent change in the microseismic regime, as with a creeping ATF up to 5 km, one would have expected a clear microseismic cloud attached to the fault up to this depth. Is it related to magnitude cutoff in the selected seismicity for the figure? This question relates to the previous one, as a locking depth deeper than 5 km may provide an acceptable or even better fit.

(5) The lateral boundary conditions looks odd to me. The text writes that the SW edge is moving to the SW at the rate of 0.5 mm/yr, whereas the NE edge has a rate of 3.5 mm/yr to the NE. This is clearly seen on Figure 5 (ATF1 and ATF3) where the Von Mises stress is uniform on the vertical edge. However, a realistic mechanical model should have a non uniform displacement rates at the NE edge, with a 3.5 mm/yr above

ATF, and some much smaller velocity beneath it (as the deepest crust there is part of the SW block). This problem is related to the specific geometry of the fault in this purely elastic modeling, which leads to unphysical boundary conditions. The way to correctly deal with this is not clear to me: what is the rate to be taken at depth beneath ATF on the NE border?If the ATF reaches the NE border precisely at the deep angle of the rectangle model, would this stabilize the problem? What could be specified as boundary conditions if the ATF reaches the horizontal, lower border? Probably, the safest way to solve this problem is to work with a viscous layer simulating the lower crust, as is usually done.

(6) The authors do not discuss the faulting process of the upper, locked part of ATF. Is this locked part seismogenicÂă? If yes, what is the expected magnitude? If one takes 5 km as the locking depth, as suggested by the authors, the shallow ATF has a width around 15 km, and a width 60 km, thus leading to a potential of a few magnitude 6.5, possibly close to 7 if it breaks in one single rupture. The slip could be in the 1.5- 2.5 m range. Thus, with a average ATF slip rate of at least 4 mm/yr, such an event should occur very roughly every 200-400 years – which is contradicted by the the historical records. This gets worse if one accepts that a deeper locking depth (7, or even 9 km) remains plausible (ie, fits equally the GPS data, as suggested above). Indeed, magnitudes of 7 and above would then become possible, and the absence of historical earthquakes could then suggest that the fault is in its latest part of its seismic cycle. Of course, alternative models may be proposed, like episodic aseismic stress release of ATF in its shallow part, which would not have occurred in the short time window the GPS monitoring – or a GPS strain rate much larger than the average interseismic one. Clearly, this is a major issue.

To conclude, my impression is that the purely elastic modeling brings a number of difficulties which makes the detailed mapping of stress and strain rate quite uncertain. This in turn makes the final conclusion, that the locking depth of the Alto Tiberina fault is about 5 km, quite uncertain as well, and more convincing tests should be done,

in particular to exclude locking depths of 7 or 9 km. The details of the calculated interaction between ATF and the secondary faults may also be questionable, due to the non physical strong mode I strain of these faults. Also, it seems that the possibilty of destructive, rare earthquakes from the locked ATF fault zone cannot be excluded.

Thus, some of the main conclusions of the paper are yet not well supported by the presented analysisÂă; the latter should integrate a more physical modeling approach, and a more carefull discussion, as suggested in more details above. This seems requested before going to a 3D modeling; the alternative being to include a standard, non-elastic rheology.

Pascal Bernard IPGP